# Genome-wide analyses reveal the contribution of somatic variants to the immune landscape of multiple cancer types

Wenjian Bi[1,2,3][ORCID], Zhiyu Xu[4][ORCID], Feng Liu[4], Zhi Xie[4], Hao Liu[4], Xiaotian Zhu[4], Wenge Zhong[4], Peipei Zhang[5,6][ORCID]*, Xing Tang[4]*

1 Department of Medical Genetics, School of Basic Medical Sciences, Peking University, Beijing, People's Republic of China, 2 Center for Medical Genetics, School of Basic Medical Sciences, Peking University, Beijing, People's Republic of China, 3 Medicine Innovation Center for Fundamental Research on Major Immunology-related Diseases, Peking University, Beijing, People's Republic of China, 4 Regor Pharmaceuticals Inc., Cambridge, Massachusetts, United States of America, 5 Department of Biochemistry and Biophysics, School of Basic Medical Sciences, Peking University Health Science Center, Beijing, People's Republic of China, 6 Key Laboratory for Neuroscience, Ministry of Education/National Health and Family Planning Commission, Peking University, Beijing, People's Republic of China

ʘ These authors contributed equally to this work.
* peipei.zhang@pku.edu.cn (PZ); tangx1986@gmail.com (XT)

**Data Availability Statement:** All relevant data are within the manuscript and its Supporting Information files.

## Abstract

It has been well established that cancer cells can evade immune surveillance by mutating themselves. Understanding genetic alterations in cancer cells that contribute to immune regulation could lead to better immunotherapy patient stratification and identification of novel immune-oncology (IO) targets. In this report, we describe our effort of genome-wide association analyses across 22 TCGA cancer types to explore the associations between genetic alterations in cancer cells and 74 immune traits. Results showed that the tumor microenvironment (TME) is shaped by different gene mutations in different cancer types. Out of the key genes that drive multiple immune traits, top hit KEAP1 in lung adenocarcinoma (LUAD) was selected for validation. It was found that KEAP1 mutations can explain more than 10% of the variance for multiple immune traits in LUAD. Using public scRNA-seq data, further analysis confirmed that KEAP1 mutations activate the NRF2 pathway and promote a suppressive TME. The activation of the NRF2 pathway is negatively correlated with lower T cell infiltration and higher T cell exhaustion. Meanwhile, several immune check point genes, such as CD274 (PD-L1), are highly expressed in NRF2-activated cancer cells. By integrating multiple RNA-seq data, a NRF2 gene signature was curated, which predicts anti-PD1 therapy response better than CD274 gene alone in a mixed cohort of different subtypes of non-small cell lung cancer (NSCLC) including LUAD, highlighting the important role of KEAP1-NRF2 axis in shaping the TME in NSCLC. Finally, a list of overexpressed ligands in NRF2 pathway activated cancer cells were identified and could potentially be targeted for TME remodeling in LUAD.

**Funding:** This research was supported by National Natural Science Foundation of China (62273010, W. B.). The funders had no role in study design, data collection and analysis, decision to publish, or preparation of the manuscript.

**Competing interests:** I have read the journal's policy and the authors of this manuscript have the following competing interests: Zhiyu Xu., F.L., Zhi Xie, H.L., X.Z., W.Z., and X.T. are employees of Regor Pharmaceuticals Inc., Cambridge, Massachusetts, USA.

## Author summary

Recent studies have found that some genetic changes help cancer cells to evade the immune surveillance. To systematically understand the impact of cancer cell genetic alterations to immune regulation, we examined 74 immune traits across 22 cancer types. The tumor microenvironment (TME), crucial for cancer development, varies based on gene mutations in different cancers. Notably, the KEAP1 gene in lung adenocarcinoma (LUAD) emerged as a key player, explaining over 10% of immune trait variations. KEAP1 mutations activate the NRF2 pathway, creating a suppressive TME in LUAD with lower T cell infiltration and heightened T cell exhaustion. Additionally, genes such as CD274 (PD-L1), associated with immune checkpoints, exhibit high expression in NRF2-activated cancer cells. By developing a NRF2 gene signature, we found that it more effectively predicts anti-PD1 therapy responses than CD274 alone in non-small cell lung cancer. Lastly, we identified ligands overexpressed in NRF2-activated cancer cells, suggesting potential targets for reshaping the LUAD microenvironment. In essence, understanding these genetic interactions helps improve lung cancer treatment and enhance the efficacy of immunotherapy.

## Introduction

Over the last decades, the discovery of immune checkpoints and their applications in cancer therapy have revolutionized the treatment of various cancer types. [1] Immune checkpoint inhibition (ICI) therapies have been utilized as single agents or in combination with chemotherapies to treat over 50 types of cancer. Despite of these tremendous success, however, only a limited percentage of patients have achieved long-lasting benefits. [2,3] The ineffectiveness of immune-oncology (IO) therapies could be at least partially attributed to the imprecise selection of patients resulted from limited understanding of tumor microenvironment (TME).

In the past, the study of cancer TME was largely restricted by the technology available for TME traits retrieval. Only a small number of TME traits can be derived from expensive and laborious experiments, such as flow cytometry [4] and immunohistochemistry [5]. Nowadays, with the advancements in omics technology and bioinformatics tools, various TME traits can be derived from RNA-seq data through de-convolution methods or gene signature enrichment analysis. [6,7] These bulk RNA-seq-derived traits have been shown to be highly consistent with immune traits obtained using cell flow cytometry or scRNA-seq. In a recent study by Sayaman et al., 139 TME traits were collected from multiple studies, mostly from bulk RNA-seq data. [8] Their results showed that germline variants account for no more than 20% of the variation in TME traits, which leaves a significant proportion of variance unexplained. [8].

Recently, the regulation of TME related to somatic alternations in intrinsic pathways in cancer cell has received increasing attention. [3,9] A large number of studies have shown that previously known oncogenes or tumor suppressors regulate the cancer TME by altering the activities of cancer intrinsic pathways. [10–12] These recurrently mutated genes in cancer cells activate or inhibit various chemokines and cytokines, resulting in different TME subtypes. Characterizing the associations between these cancer cell genetic alterations and TME traits can help to better stratify patients by TME subtypes, which could be crucial for IO therapy development.

In this study, we analyzed 74 TME traits and 22 cancer types in The Cancer Genome Atlas (TCGA), using genome-wide gene-level association analyses, to identify genes in which somatic variants significantly alter TME traits. Totally, 451 significant gene-trait associations

were reported across different cancer types. Among these associations, 14 genes were found to regulate 3 or more TME traits, which suggests their important roles in shaping multiple aspects of the TME. Of these, KEAP1 was identified as the top hit in lung adenocarcinoma (LUAD) and explained a large proportion (>10%) of variance for multiple immune traits, including interferon pathway, MHC class II expression, and the NK cell signature score. Other important gene-trait associations included TP53 mutations associated with B cell receptor (BCR) function in breast cancer (BRCA), PBRM1 mutations associated with neutrophil in kidney renal clear cell carcinoma, IDH1 mutations associated with lower lymph vessel signature in Brain Lower Grade Glioma (LGG) and BRAF mutation associated with NK cell and macrophages in Thyroid Carcinoma (THCA).

To validate our findings and further characterize the underlying mechanism, we collected and analyzed three scRNA-seq datasets of LUAD and confirmed the immunosuppressive role of KEAP1 mutations. [13–15] We found a NRF2 gene signature activated in KEAP1-mutated samples. This signature was significantly correlated to lower T cell infiltration, higher macrophage and monocyte, higher Treg percentage, and higher T cell exhaustion, indicating an inflamed but immune suppressive TME subtype. Interactions of several ligand-receptor pairs between cancer cells and immune cells (CD274:PDCD1, FAM3C:PDCD1, PVR:TIGIT) are predicted to be enhanced in NRF2 signature high samples which could be potential targets to inhibit to remodel the TME. Furthermore, using clinical trial data [16], we found that non-small cell lung cancer (NSCLC) patients with an activated NRF2 signature are more likely to experience a durable response compared to other patients (p-value: 0.018). By contrast, CD274, a biomarker that has already been adopted for use in lung cancer clinical traits (NCT04294810) [17,18], failed to separate responsive patients from others (p-value: 0.51).

Comprehensive genomic analyses have identified somatic mutations and other alterations in the *KEAP1* or *NRF2* genes in various types of cancer, and Nrf2 mutations occur less frequently than Keap1 mutations. [19] Disruptions in the Keap1-Nrf2 pathway is frequently associated with poor prognosis and chemotherapeutic resistance in NSCLC. [20] Since the Keap1-Nrf2 pathway plays as primary regulator of key cellular processes that contribute to resistance against chemotherapy drugs, NRF2 has been studied as a potential therapeutic target molecule in NSCLC and some other cancers. [21] Our findings explore the molecular features and the impact of KEAP1-NRF2 to TME, which will be beneficial for novel treatment approaches in NSCLC in the near future.

## Results

### Overview of genetic test to associate somatic variants with TME traits

We conducted genome-wide gene-level association analyses to identify genes in which somatic variants alter TME traits significantly. Of the TME traits Sayaman et al. analyzed [8], we selected 74 traits (S1 Table), most of which were derived from bulk RNA-seq data of TCGA tumor samples by scoring different gene signatures using ssGSEA or by deconvoluting bulk RNA-seq using CIBERSORT. [8,22] These immune traits were selected to represent proportion of different immune cell types, activity of immune pathways, and states of different immune cells. To ensure data processing workflow consistency across different cancer types, we used somatic mutations, log2 copy number alterations, and clinical data from TCGA pan-Cancer project. [23] In total, 22 cancer types with > 100 samples were selected for analysis (S2 Table). Somatic mutations in non-coding regions (UTR or intron) except splicing changing variants were excluded from analysis. For each cancer, genes mutated in ≥ 5 tumor samples were fed into association test pipeline. Immune traits were transformed to either quantitative or binary values depending on their distributions, and then passed to linear or logistic

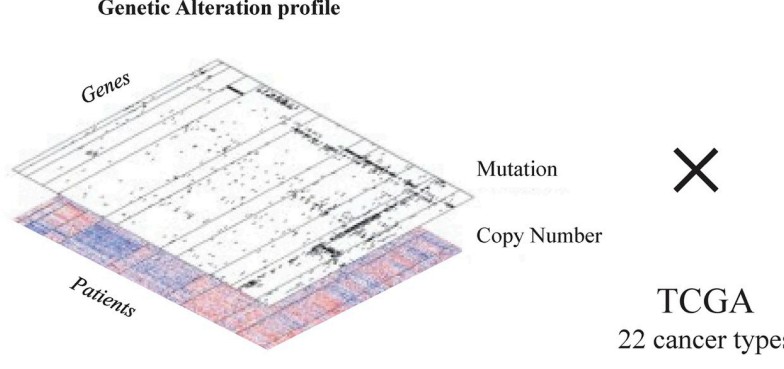

**Fig 1. Overview of genetic association test to identify immune regulators in different cancer types using TCGA data.** Somatic mutations and copy number variations from 22 cancer types are downloaded from cbioportal to associate with 74 immune traits collected from Sayaman et al. (2021) [8].

regressions (see Methods). In addition to the confounding covariates adjusted by Sayaman et al. (including gender, days to birth, and age at initial pathologic diagnosis), radioactive therapy status and chemotherapy status were also incorporated in the regression model [8].

Both somatic mutations and copy number alterations (CNA) can impact cancer via TME regulation. However, CNA usually spans multiple gene regions and thus it is challenging to distinguish a driver gene from other passengers in the same CNA region without prior knowledge or additional experiments. Hence, we mainly focused on associations between mutations and TME traits, and included CNA as an additional confounder. Although both quantitative log2CNA value and categorized CNA value are available in cbioportal [24], we incorporated log2CNA as a covariate since it has higher correlation with gene expression. The log2CNA values were adjusted by tumor purity prior to analysis (see Methods).

In the germline variants association study, Sayaman et al. combined multiple cancer types together for analysis. [8] In this study, we analyzed 22 cancer types separately due to a significant diversity of mutations in cancer cells. A total of 1,628 (74×22) genome-wide analyses were conducted for 22 cancers and 74 TME traits (Fig 1). To avoid a large number of spurious positives, analyses p-values were adjusted using genome control lambda (gc lambda, see Methods). For each genome-wide analysis, we calculated false discovery rate (FDR) adjusted p-values as q-values and selected genes whose q-values < 0.05 as top genes that are significantly associated with TME traits.

## Contribution of somatic variants to TME traits varies across cancer types

We identified 451 significant gene-trait associations with q-values < 0.05 across 22 cancer types (S3 Table). For cancer types of brain Lower Grade Glioma (LGG), Thyroid Carcinoma (THCA), and Breast Cancer (BRCA), 99, 81, and 62 significant gene-trait associations were identified, respectively. Fig 2A demonstrated the 14 genes significantly associated with 3 or more TME traits, of which each of the 13 genes except TP53 was identified in only one corresponding cancer type, suggesting the importance of tissue context on TME regulation. Based on the identified gene-trait associations, we calculated the proportion of trait variance that is explained by somatic variants, including both somatic mutations and CNA, for each cancer type (see Methods). Totally, 32 TME traits from 9 cancer types have >20% variance explained by somatic variants, suggesting a non-negligible contribution of somatic variants to TME (Fig 2B and S4 Table).

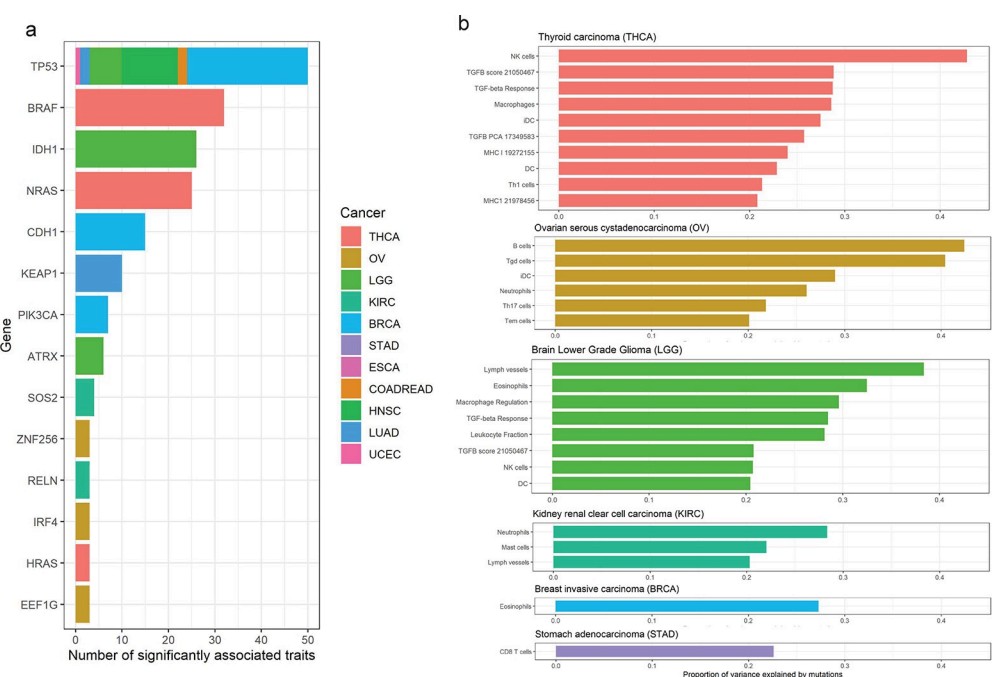

**Fig 2. Different TME characteristics of different cancers are shaped by differentially mutated genes. (a). Top genes that are significantly associated with > = 3 TME traits.** Of the 451 significant gene-trait associations, 14 genes were highlighted. The y-axis is the gene name and the x-axis is the number of the traits that are significantly associated with the gene. Different colors are for different cancer types. **(b). Immune traits with > 20% explained variance by somatic mutations**. The y-axis is the name of TME trait and the x-axis is the proportion of the variance explained by somatic mutation. Different colors are for different cancer types.

## TP53 regulates TME traits for multiple cancer types

As aforementioned, TP53 is the only gene associated with multiple TME traits (> = 3) in multiple cancer types (Fig 2A). In BRCA, TP53 mutations are correlated to higher tumor-infiltrating lymphocyte (TIL) fraction, higher interferon gamma, higher macrophage, and higher B cell receptor (BCR) richness and diversity (S5 Table), which is consistent with previous results that TP53 mutations may promote immunogenic activity in BRCA. [25] On the contrary, in head and neck squamous cell carcinoma (HNSC), TP53 mutations are associated with inhibitory immune features such as lower lymphocyte infiltration signature score and lower CD8 cell signature score, which was also reported in previous studies. [26] The example of TP53 highlights the importance of tissue context for TME regulation, emphasizing the need of cancer-type-specific TME stratification for targeted immunotherapy.

## Kidney Renal Clear cell carcinoma (KIRC)

In KIRC, we identified neutrophils signature, for which, 28% trait variance are explained by somatic variants (Fig 2B). Neutrophils were known as the first line of defense against microbial infection. They circulate in the blood and are recruited rapidly to the site of tissue injury. Recent studies showed that neutrophils have pro-tumoral or anti-tumoral functions under different contexts. For KIRC specifically, previous studies suggested that tumor-infiltrating neutrophils act as an independent adverse prognostic feature. [27] Higher tumor-infiltrating neutrophils (TINs) were significantly associated with worse overall survival and higher metastasis rate.[27]

Of the genes constituting the trait of neutrophils signature in KIRC, we found PBRM1 contributes mostly to the trait variance (~22% variance explained, S5 Table) and mutations of PBRM1 are associated with higher neutrophils (p value: $1.95 \times 10^{-9}$). This perhaps is not surprising as PBRM1 encodes a protein that is involved in the regulation of chromatin remodeling and inflammation-related genes are highly regulated by chromatin remodeling genes in KIRC. [28] PBRM1 is highly mutated (~30%) in KIRC samples in which more than 85% of the genetic alterations lead to loss of function (deletion, truncating mutation, and splice mutation). PBRM1 mutation cause activation of inflammation-related genes, which can trigger neutrophil-dependent lung metastasis in advanced KIRC. [28] Generally speaking, experiment and analyses in both mice and human validate that PBRM1 loss of function defines a nonimmunogenic tumor phenotype associated with checkpoint inhibitor resistance in renal carcinoma. [28,29]

### Brain Lower Grade Glioma (LGG)

In LGG, we identified lymph vessels, for which, 38% trait variance are explained by somatic variants. Of the genes significantly associated with the lymph vessels trait, IDH1 gene contributed mostly to the trait variance (33% variance explained). IDH1 is highly mutated in LGG tumor samples (77%) in which almost half of the mutations are annotated as truncating mutations. IDH1 mutations are associated with lower lymph vessel signature (p-value: $7.97 \times 10^{-40}$). IDH1 is reported to regulate podoplanin (PDPN) expression in glioma by regulating its promoter methylation status. [30] And PDPN is strongly expressed in higher-grade IDH1-wild-type glioma but almost undetectable in IDH1-mutated glioma. Consistent with our analysis that IDH1 loss of function mutations is associated with lower lymph vessel signature, upregulation of PDPN induces lymphangiogenesis and metastasis in tumor. [31]

### Thyroid Carcinoma (THCA)

For THCA, somatic variants account for 42.8% variance of NK cells, 28.8% variance of TGFB score, 28.6% variance of macrophages, and 22.9% variance of dendritic cells (DC). They are mainly driven by *BRAF* gene mutations which account for 17.8% - 39.5% of variance for the four traits. In THCA, 56% of patients are carrying the same mutation BRAF V600E mutation which is almost the only mutation found in *BRAF*. Based on our analysis, BRAF V600E mutation is significantly associated with high macrophages (p-value: 2.42e-15). In addition, patients carrying the BRAF mutation are having lower CD8 T cells % (p-value: 4.00e-7) and higher Treg cell % (p-value: 6.64e-12), which is consistent with another study of human thyroid cancer.

### Breast Cancer (BRCA)

Besides TP53, another major influencer to TME traits in BRCA is CDH1. Most of CDH1 mutations in TCGA-BRCA cohort are loss of function mutations, of which > 80% are from LumA subtype. Deficiency of CDH1 protein is associated with higher NK cell signature score and lower macrophage cell population. This is in line with its function of encoding a ligand for interacting with killer cell lectin-like receptor G1 (KLRG1) on NK cells and memory T cells to trigger inhibitory signals. [32] It is interesting that CDH1 loss of function (LOF) mutations are highly frequent in LumA breast cancer, suggesting that high expression of LOF CDH1 mutants in LumA cancer indicates worse prognosis. Furthermore, deficiency of CDH1 maybe oncogenic for cancer initiation and over expression of CDH1 mutants could advance cancer progression by inhibiting NK cells and T cells. Several studies have shown synergistic effect by blocking KLRG1 and PD-1 together in mouse models for multiple cancer types. [33, 34] Our

analysis supports the notion that CDH1-high BRCA patients can be potentially treated by a combination therapy of KLRG1 inhibitor and PD-1/PD-L1 inhibitor.

## Lung Adenocarcinoma (LUAD)

In LUAD, KEAP1 stands out as the top hit associated with multiple TME traits, explaining 5% to 15% trait variance (Fig 3A). Boxplot of trait value distribution shows that many immune traits are downregulated in KEAP1 mutated patients, such as TGFβ signaling, different T cell subtypes gene signatures, NK cell gene signature, MHC expression, macrophage gene signature, and interferon pathways (Fig 3B). Per our interest of LUAD, we used scRNA-seq data to further confirm these associations and characterize the molecular mechanisms behind the associations.

## NRF2 pathway activation in cancer cell shapes suppressive TME in LUAD

KEAP1 is an adaptor protein connecting target protein to CUL3 (Culin 3)/RBX1 (Ring box 1) E3 ubiquitin ligase complex for protein degradation. NRF2 is known as a transcription factor driving expression of antioxidant genes. KEAP1 constitutively targets NRF2 for proteasomal degradation, thereby prevents nuclear accumulation of NRF2 and the downstream activation of the antioxidant gene expression. [35] It is known that most somatic mutations in KEAP1 are missense or truncating events that can generate dominant negative forms of KEAP1. [36] Somatic mutation of KEAP1 causes deficiency of KEAP1 function, which in turn reduces degradation of NRF2 protein, resulting in activation of NRF2 downstream genes. To validate KEAP1's regulation of NRF2 pathway in LUAD tumor, we curated a NRF2 target gene list by

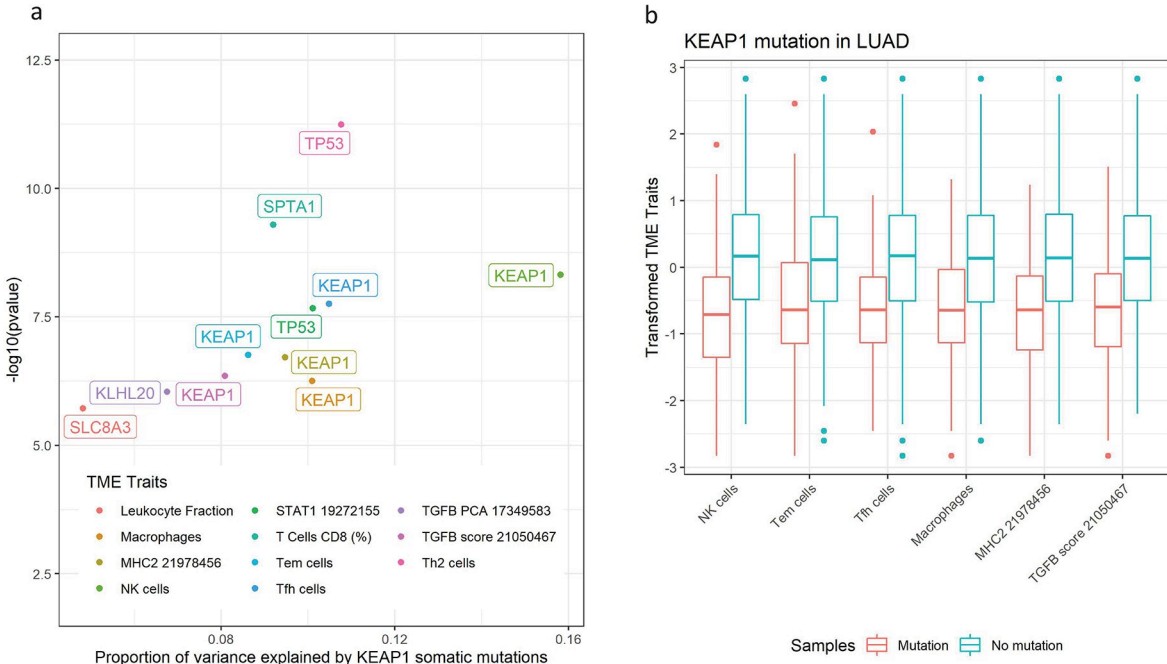

**Fig 3. KEAP1 is the gene shaping multiple TMEs in LUAD. (a). Immune traits that are associated with KEAP1 mutations in LUAD.** The y-axis is -log10(p values) in which the p value correspond to the tests associating somatic mutation in KEAP1 with TME traits. The x-axis is the proportion of trait variance been explained by KEAP1 mutations. **(b). Immune traits value distribution across KEAP1 mutated group (red) and KEAP1 wild type group (blue) in LUAD**. The x-axis is for different TME traits and the y-axis are for TME traits after transformation. The p-values in the x-axis are calculated using two-sample students t-test.

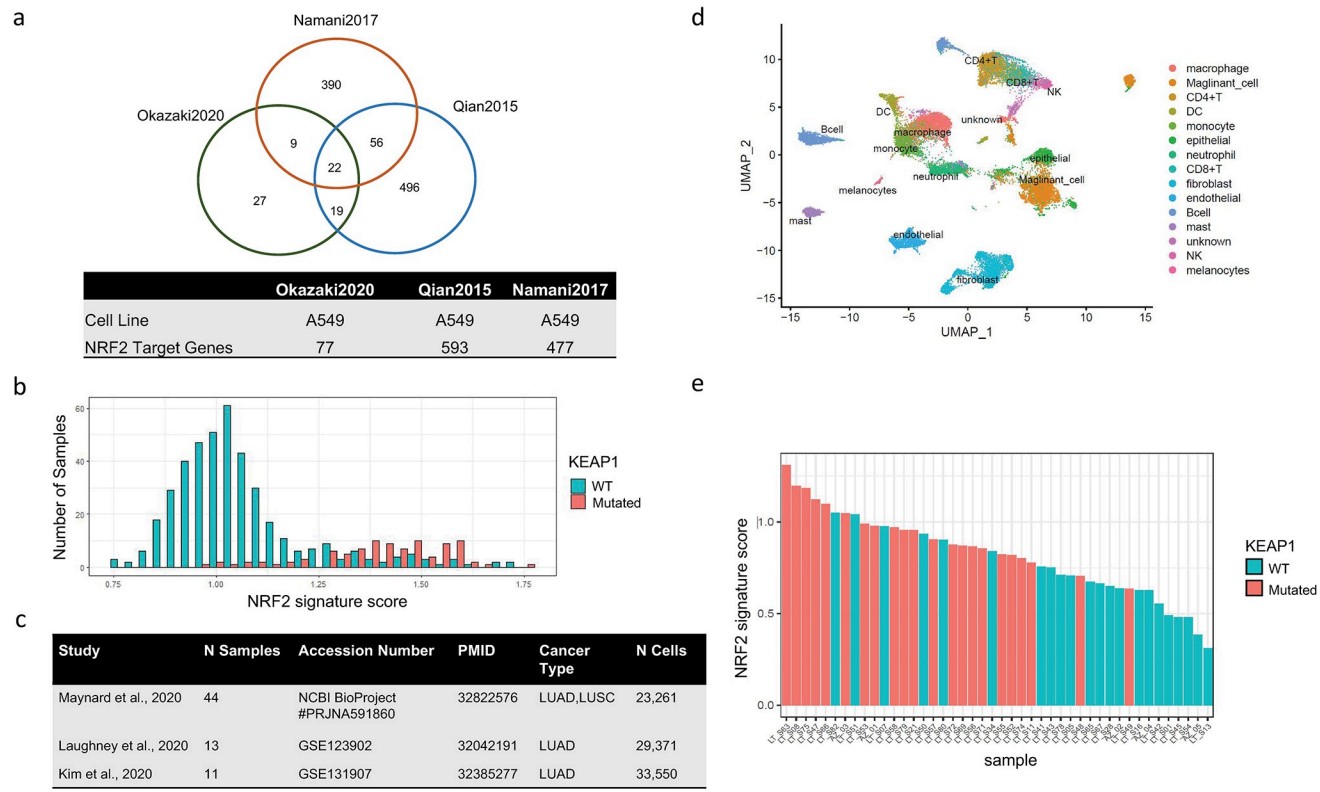

**Fig 4. Identification of NRF2 gene signature and its association with KEAP1 mutation status in both bulk tumor RNA-seq setting and scRNA-seq setting.** (a). Venn diagram for gene lists collected from three independent studies and their associated information [13–15]. (b). Distribution of NRF2 gene signature score for tumor samples from TCGA LUAD cohort colored by KEAP1 mutation status. One-sided Wilcoxon test p value < 2.2e-16. (c). Collected single cell RNA-seq datasets for validating findings from TCGA data and characterizing mechanism of immune regulation. (d). UMAP plot with annotated cell types for Maynard et al., 2020 study [13]. (e). Distribution of NRF2 gene signature score for cancer cells of tumor samples from Maynard et al. scRNA-seq study colored by KEAP1 mutation status. Fisher exact test p value = 0.00604 [13].

integrating results from 3 independent studies. [37–39] Each of the studies has reported a NRF2 downstream gene list by NRF2 knock down or overexpression experiment in lung cancer cell lines. To minimize the impact of noise from each single study, we included only target genes that are supported by at least 2 studies into the NRF2 gene signature (Fig 4A). NRF2 gene signature score was then calculated for each sample in TCGA lung adenocarcinoma cohort by ssGSEA. We found a good separation of NRF2 gene signature score distribution between KEAP1 mutated group and KEAP1 wild type group (wilcox single sided test pvalue < 0.05), suggesting that KEAP1 mutation drives activation of NRF2 pathway in LUAD tumor (Fig 4B).

To further validate mutation-trait associations, we did a systematic search for whole tumor scRNA-seq data for lung adenocarcinoma from public domain and got 3 high quality datasets (Fig 4C). [13–15] Raw counts of corresponding studies were downloaded and processed using the recommended parameters by each study. We accepted the annotation of cancer cells from each study but re-annotated immune cells with a standard pipeline. Given that the Maynard et al study (n = 44) (Fig 4D) has the largest sample size and also has mutation status of KEAP1, we used this dataset for primary analysis and used the other two study datasets for validation [13]. NRF2 gene signature score calculated from cancer cell pseudo-bulk expression separates KEAP1 mutated samples from non-mutated samples in Maynard et al study, suggesting that NRF2 gene signature is a good indicator of KEAP1 mutation status (Fig 4E, fisher exact test p

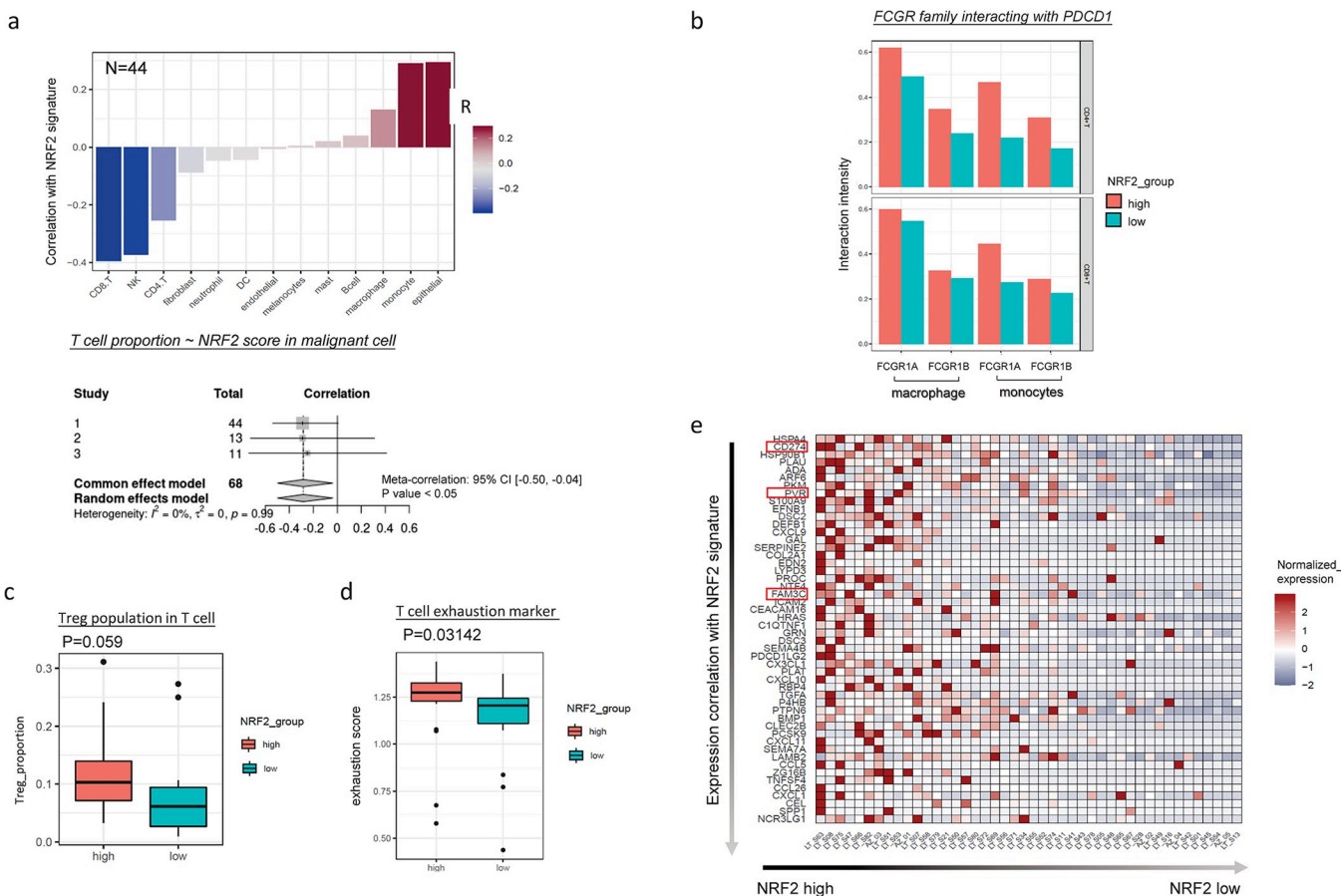

**Fig 5. Associations between NRF2 signature score and different immune traits in Maynard et al. scRNA-seq study [13].** (a). Correlation between NRF2 signature score and various immune cell types percentage within Maynard et al. scRNA-seq study (upper panel) [13]. Red indicates positive correlation while blue indicates negative correlation. Forest plot of meta analysis for correlation of NRF2 signature score with T cell proportion across 3 independent scRNA-seq datasets(lower panel). The fixed effects model was applied. The grey squares represent the correlation coefficient of each dataset; the different size of the squares reflect the weight of each study in the meta-analysis. The horizontal lines indicate the 95% confidence intervals of each study. The diamond represents the effect size (pooled correlation coefficient). (b). CellphoneDB predicted interactions score between FCGR family (FCGR1A and FCGR1B) from macrophage/monocyte and PD1 from T cell. Tumor samples are classified by by NRF2 signature score (NRF2high, NRF2low). (c). Comparison of Treg population across samples classified by NRF2 signature score (NRF2high, NRF2low). (d). Comparison of T cell exhausion across samples classified by NRF2 signature score (NRF2high, NRF2low). (e). Gene expression heatmap of the top 50 ligands that are positively correlated with NRF2 gene signature in malignant cells. For each sample, pseudo bulk is aggregated from all malignant cells and calculate the averaged expression of ligands and NRF2 gene signature from pseudo bulk data. To study ligands that are highly associated with NRF2, we performed correlation analysis for each ligand with NRF2 gene signature. In the plot, row represents ligands ranked by its correlation with NRF2 signature (top to bottom: highest to lowest). Column represents tumor samples that are ranked by NRF2 gene signature (left to right: highest to lowest). Colors in the heatmap represent the averaged expression of each ligand in pseudo bulk data that are normalized across different samples.

value < 0.05) and we could use NRF2 gene signature to infer the molecular impact from KEAP1 mutations in other scRNA-seq studies for which no mutation information is available [13]. By correlating NRF2 signature score with various immune cell types, we found that whereas it is negatively correlated with T cell population from all three datasets (Fig 5A), it is positively correlated with macrophage and monocytes population. Although previous studies described that solid tumors contain a significant population of tumor-infiltrating myeloid cells, which promote tumor growth by suppressing the immune system, the contribution of myeloid cell to cancer progression is quite complicated. [40] Using in vivo imaging, Arlauckas et al. showed that the spatial association between macrophages and CD8 T-cell was responsible for resistance to anti-PD1 therapies. [41] Consistent with the report, we also found enhanced

interactions between FcγR genes from macrophage/monocytes and PD1 from T cells in NRF2 Signature high samples predicted by CellPhoneDB [42] from Maynard dataset (Fig 5B).

Although NRF2 signature is negatively correlated with T cell population, it is positively correlated with Treg cell population, which suggests that NRF2 activation in cancer cells may promote Treg differentiation and maturation (Fig 5C). The p-value 0.06 is quite marginal possibly due to small sample size (n = 44). Since Treg cells are involved in inducing T cell exhaustion, we further associated NRF2 signature with T cell exhaustion markers derived from Caushi's study. [43] NRF2 signature high samples correspond to higher T cell exhaustion score (Fig 5D, p-value: 0.03). To better characterize cell-cell communication changes due to NRF2 pathway activation, we plotted out the expression profile of top 50 ligands positively correlated with NRF2 signature in cancer cells (Fig 5E), which indicated that three genes (CD274, PVR, FAM3C) are ligands for well-known inhibitory immune check point receptors PDCD1 and TIGIT on T cells. CellPhoneDB predicted interactions between these immune checkpoint ligand receptor pairs are enhanced in NRF2 signature high samples (Fig 6A). While CD274 expression is highly correlated with NRF2 signature score in LUAD cancer cells (Fig 6B), chromatin immunoprecipitation (ChIP) experiment in NRF2 activated human primary melanocytes confirmed direct binding of NRF2 to CD274 promoter, [44] suggesting that NRF2 may activate CD274 expression directly at transcription level. For other inhibitory ligands whose expression is highly associated with NRF2 signature, we found that PVR expression is associated with worse survival in LUAD patient cohort significantly (Fig 6D, p.value < 0.01). Expression of FAM3C also shows trend of association with worse survival in LUAD patients after 50 months although the p value is not significant due to the fact that only small number of patients live longer than 50 months (Fig 6C, p.value = 0.16). We report these findings to the field to support new immunotherapy target identification for treating these NRF2 pathway activated LUAD patients.

Our analysis confirmed the immune suppressive role of NRF2 pathway activation in LUAD and identified potential signaling transduction molecules from cancer cells to immune cells for conducting the suppressive regulatory function. To gain a more comprehensive view about intrinsic pathway changes within KEAP1 mutated cancer cells, we performed differential gene expression analysis with pseudo bulk gene expression profile of aggregated cancer cells and followed by GSEA analysis using the Maynard dataset. We found out that NRF2 activated genes are enriched in well-known immune regulation pathways such as interferon pathways, TGFβ pathway, TNFα pathway and complement pathway (Fig 6E). These pathways mainly mediate inflammatory cytokine synthesis and secretion and establish a cancer-prone inflammatory microenvironment to promote lung cancer progression.

## NRF2 signature indicates better prognosis for anti-PD1 or anti-PD-L1 therapies

We proposed a model to summarize our findings for KEAP1-NRF2 regulation of TME (Fig 7A). Mutation of KEAP1 causes loss of function of KEAP1, which prevents NRF2 protein from degradation. Accumulation of NRF2 in cancer cells may activate expression of inhibitory cytokines and ligands directly by transcriptional regulation or by activating inflammation pathways such as interferon pathways, TGFβ pathway, TNFα pathway, and complement pathway. These inhibitory ligands or cytokines expressed or secreted from cancer cells inhibit cytotoxic T cell infiltration directly or induce T cell exhaustion which facilitates formation of a suppressive cancer TME type. This TME type defines a unique LUAD patient cohort with worse survival status (Fig 7B). The upregulation of CD274 (PD-L1), FAM3C, and other T cell exhaustion markers in NRF2 activated LUAD cancer suggest that they could be the population who can benefit the most from anti-PD1 or anti-PD-L1 therapy. To test this hypothesis, we

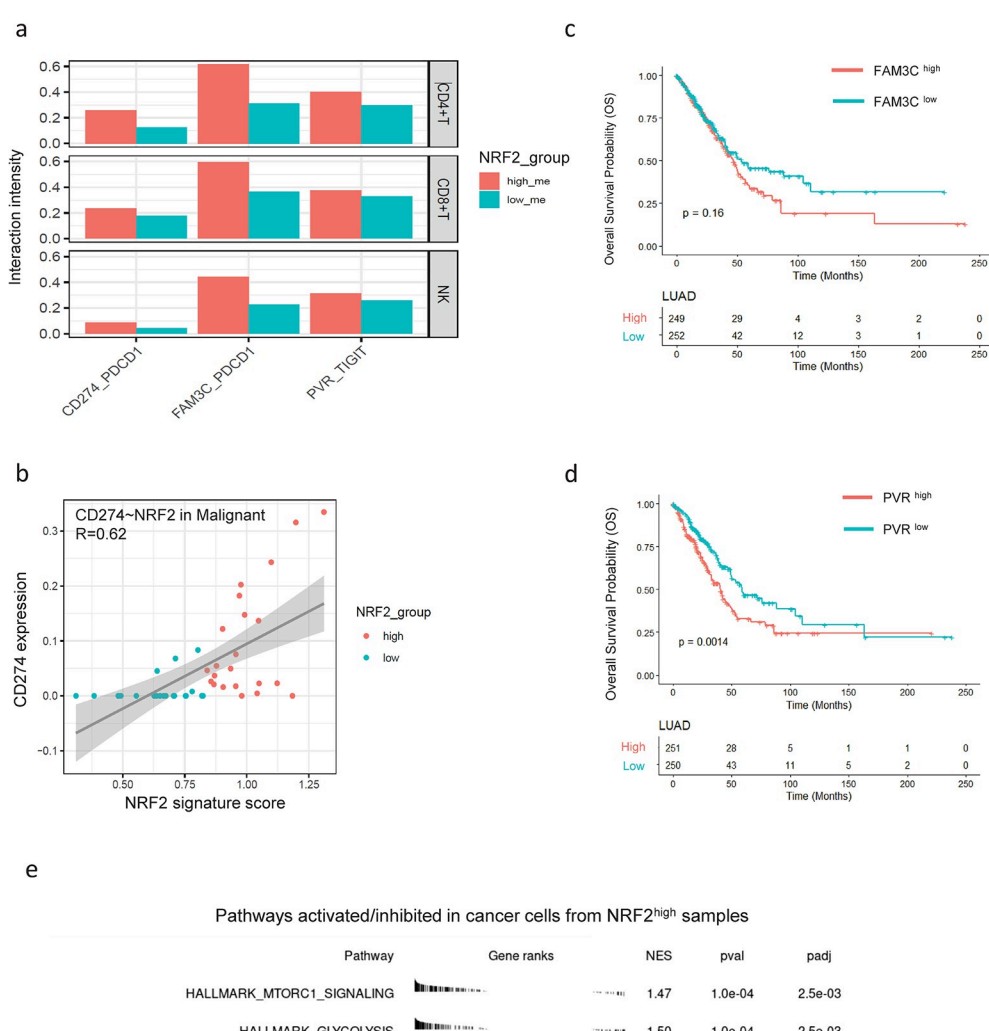

**Fig 6. Mechanisms of KEAP1-NRF2 pathway in promoting suppressive TME.** (a). Interactions predicted by CellphoneDB of immune checkpoint ligand-receptor pairs between cancer cell (ligand: CD274, FAM3C and PVR) and multiple immune cell types including CD4+ T cell, CD8+ T cell and NK cells) (receptor: PD1 and TIGIT). (b). Correlation of CD274(PD-L1) gene expression and NRF2 signature score within malignant cells (patient samples are classified by NRF2 signature). (c). Survival analysis of FAM3C in TCGA LUAD patient cohorts. Patients are stratified based on gene expression. We observe a few individuals with a long follow-up record. The examples include sample of TCGA-49-AARQ censored at around 224 months, sample of TCGA-78-7163 censored at around 242 months, and sample of TCGA-78-8640 censored at around 235 months. (d). Survival analysis of PVR in TCGA LUAD patient cohorts. Patients are stratified based on gene expression. (e). Enriched pathways for differential expressed genes between malignant cells of NRF2 high samples and NRF2 low samples.

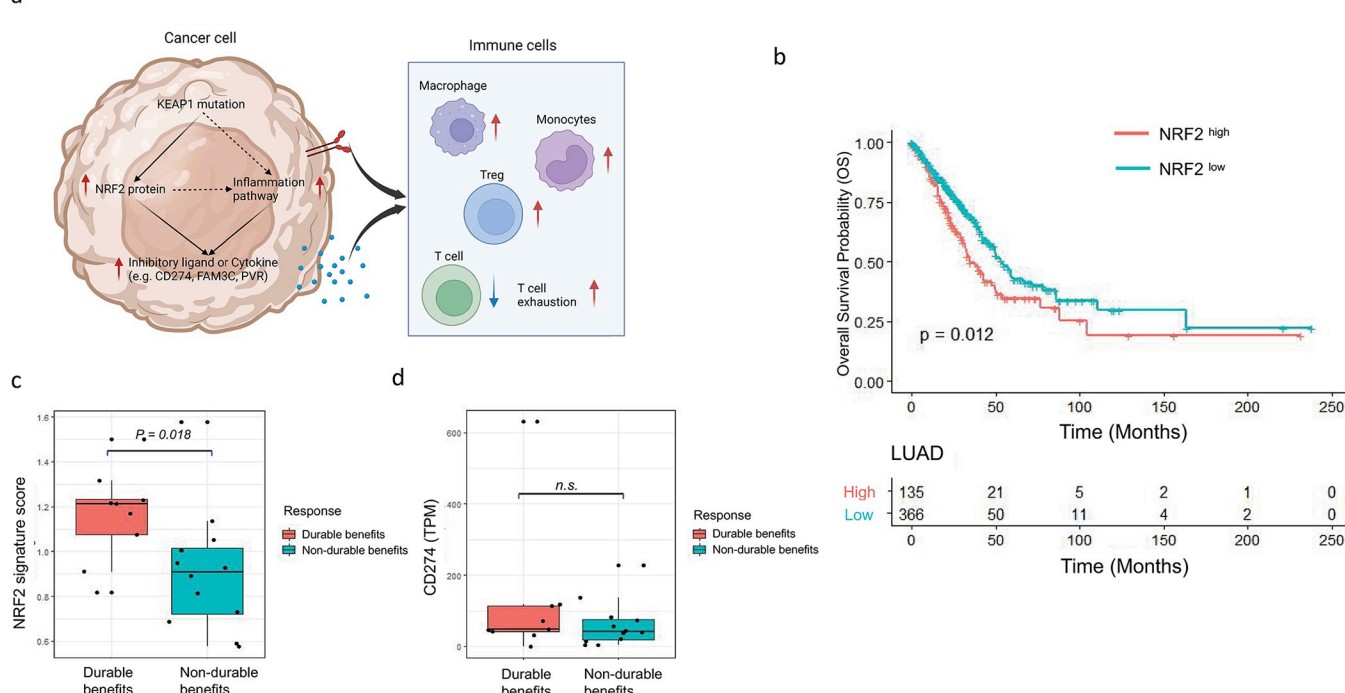

**Fig 7. Summary of KEAP1-NRF2 regulation to LUAD TME and its clinical application.** (a). Proposed model for KEAP1-NRF2 regulation to TME in LUAD. (b). Survival curve of TCGA LUAD cohort grouped by NRF2 signature score (log-rank p value: 0.012). For TCGA datasets, 1.25 is selected as the cutoff to stratify patient samples into NRF2high and NRF2low. (c). Boxplot of NRF2 signature score of tumor samples before anti-PD1 treatment from durable benefited patient group and non-durable benefited group. Wilcox test shows significantly higher NFR2 signature score in tumor samples collected from patients with durable benefits for anti-PD1 treatment (p value = 0.018). (d). Boxplot of CD274 gene expression of tumor samples before anti-PD1 treatment from durable benefited patient group and non-durable benefited group.

analyzed public anti-PD1 clinical trial data. [16] The patients with durable benefits are having higher NRF2 signature score comparing to non-durable benefits group (Fig 7C, wilcox test p-value: 0.018). Surprisingly, there is no significant difference in CD274 (PD-L1) expression between the two patient groups (Fig 7D), suggesting that NFR2 gene signature potentially can be used as a superior predictive biomarker over CD274. However, sequencing a single gene is much easier than doing expression quantification of a panel genes. Since KEAP1 mutations cause upregulation of NFR2 and its downstream signature genes which is confirmed in our analysis (Fig 4B), KEAP1 gene sequencing can be a more applicable test to be used in clinical setting than NFR2 gene signature quantification by either RNA-seq or microarray.

## Discussion

In this study, we systematically investigated the contribution of somatic variants to TME traits using TCGA cancer data. The results demonstrated that somatic variants play an important role in shaping cancer TME in tissue-specific and cancer-type-specific manner. A total of 451 significant gene-trait associations were identified across 22 cancer types, with 14 genes significantly associated with three or more TME traits, including IDH1 in KIRC, BRAF in THCA, CDH1 in BRCA, and TP53 in multiple cancer types.

We select KEAP1-NRF2 in LUAD as an example to highlight the value of our study in patient stratification for IO therapies. Our results showed that KEAP1 mutated or NRF2 activated LUAD samples share unique TME characteristics, such as lower T cell infiltration, higher T cell exhaustion level, and higher expression of immune checkpoint ligands that are targeted

by existing therapies. Real data analysis revealed that the NRF2 gene signature curated in our study could serve as a better biomarker than the currently used biomarker CD274 for selecting patients for anti-PD1/PD-L1 therapies. Moreover, we observed upregulation of the PVR gene (ligand for TIGIT) in NRF2 signature high LUAD tumors, which suggests that targeting NRF2 signature high population with anti-PD1 (or anti-PD-L1) and anti-TIGIT combo therapy may achieve synergistic effect. By May of 2022, the phase 3 SKYSCRAPER-01 study (NCT04294810) evaluating the addition of tiragolumab (anti-TIGIT) to atezolizumab (anti-PD-L1) as first-line treatment for people with PD-L1-high, locally advanced or metastatic non-small cell lung cancer (NSCLC) did not meet its co-primary end point of progression-free survival (PFS), according to Roche (https://bit.ly/37EbDJX). We propose that NRF2 signature can be used to identify patients that may benefit from tiragolumab and atezolizumab combination therapy more precisely than CD274 gene only. The NRF2 gene signature reported in our study could serve as a biomarker to define a patient group with specific TME subtype which may benefit the most from immune-therapy.

In addition to KEAP1-NRF2, other key genes reported in our list (Fig 2) may also be involved in shaping specific TME type. TP53 mutation defines an immunogenic and lympho-cytes infiltrated TME type in breast cancer and defines a low cytolytic T cell infiltration TME type in head and neck squamous cell carcinoma. Loss of function mutation in chromatin remodeling gene PBRM1 activates inflammation gene expression and defines a nonimmuno-genic and neutrophil-rich TME type in KIRC which is metastatic and irresponsive to immune check point therapy. IDH1 mutation in brain lower grade glioma associates with lower lym-phangiogenesis and metastasis possibly by down regulating PDPN gene expression. CDH1 mutation in breast cancer defines a NK cell infiltrated/activated TME type by releasing the inhi-bition from NK cell inhibitory receptor KLRG1. We noticed that gene mutations in cancer cells not always promote immune evasion. To the opposite, it's quite often that cancer cell mutations such as in IDH1 and CDH1 enhance immunity. Mutations in these genes have oncogenic effect in cancer cells but can also promote immunity at the same time. Cancer cell mutations play dynamic and multi-layer regulatory roles for cancer development and progression. The somatic mutations can result in TME traits variation and be used to define TME subtypes. The associa-tions between cancer cell genetic alterations and TME traits reported from our study provide unbiased evaluations of TME contributions from cancer cell genetic alterations, which can help stratify patients and allow researchers to develop specific targeted IO therapies.

In the main text, the genome-wide analyses mainly focused on coding regions and treated the somatic mutations equally. We noticed the research studies that highlighted the important role of noncoding RNAs in remodeling TME. [45,46] Hence, we also conducted multiple sen-sitivity analyses in which 1) somatic mutations in non-coding regions were included; 2) only somatic mutations classified as missense or nonsense were included; and 3) driver somatic mutations were double-weighted. The definition of the driver somatic mutation is from OncoVar [47], an integrated database and analysis platform for oncogenic driver variants in cancers. The detailed results can be found in S1 Fig and S6–S9 Tables. In general, upweighting driver mutations does not significantly help find more findings. If the analysis targeted Mis-sense Mutation only, a significant majority of the initially identified signals were successfully replicated. That is because Missense Mutation takes more than 52.9% of the somatic mutations (S10 Table). Meanwhile, when the analysis confined to Nonsense Mutations exclusively, only a limited number of the initially identified signals were reproduced because Nonsense Mutation only takes ~ 4% of the somatic mutations (S10 Table). And both analyses do not exclusively identify many signals missed by the initial analyses. Incorporating the somatic mutations in non-coding regions revealed a slightly broader spectrum of findings, of which potentially sig-nificant discoveries include UBTF in Ovarian Cancer (OV) and ACTBL2 in HNSC (S6 Table).

Please note that all conclusions have been drawn from bioinformatic analyses. We fully acknowledge the importance of experimental validation as a necessary step for future research.

## Materials and Methods

### TCGA source and data transformation

Genetic alteration data was downloaded from https://www.cbioportal.org/ for each cancer type. Clinical data was downloaded from https://portal.gdc.cancer.gov/. Immune traits were downloaded from Supplementary Tables 2–3 of Sayaman et al.[8]

We first transform the TME traits to a quantitative or binary value depending on its distribution (S2 Table). For a TME trait, 1) if the raw trait values of more than 50% subjects are 0, then we dichotomize the trait to 0 and 1, depending on if the raw value is 0 or not; 2) if the raw trait values of more than 10% subjects are 0, we dichotomize the TME traits to 0 and 1, depending of if the raw value is less than median or not; 3) otherwise, we use inverse normalization transformation to calculate a quantitative value.

Based on Variant Classification from cbioportal, we excluded somatic mutations annotated as 3'Flank, 3'UTR, 5'Flank, 5'UTR, Intron, and RNA. For each gene, if the sample is a somatic mutation carrier, the genotype was coded as 1, otherwise, the genotype was codes as 0. Genes with fewer than 5 somatic mutation carriers were excluded from analysis.

The basic idea of the transformation follows previous studies of Sayaman et al, [8] where parts of the TME traits were also dichotomized and treated as binary variables in case of a large number of 0 values. For example, for ~ 50% of the participants, the corresponding TME traits of B Cells Memory (%) are 0. In this case, the trait cannot be normalized, as requested by linear regression. The data transformation process is completely data-driven and remains the same for all TME traits and cancer types.

### Genome-wide gene-level association testing

We use linear and logistic regression to associate the transformed TME traits and gene-level somatic mutations. Potential confounders of gender, days to birth, age at initial pathologic diagnosis, radiation therapy, and chemotherapy history were incorporated if applicable (S1 Table). For each gene, we also incorporated the corresponding log2CNA value in the model after adjusting for tumor purity as fomula 3 which is derived from fomula 1 and 2 by assuming 2 copy of genes in normal cells.

$$\text{Log2CNV}_{\text{ratio}} = \text{log2CNV}_{\text{tumor}} - \text{log2CNV}_{\text{normal}} \qquad \text{Formula 1}$$

$$\text{CNV}_{\text{tumor}} = \text{CNV}_{\text{CancerCell}} * \text{TumorPurity} + 2*(1 - \text{TumorPurity}) \qquad \text{Formula 2}$$

$$\text{adjusted log2CNA} = \log2(2^{\alpha} - 2*(1 - \text{purity})/\text{purity}) - 1 \qquad \text{Formula 3}$$

where $\alpha$ = raw log2CNA + 1

TME traits are complicated, and the confounders mentioned above can only cover a limited proportion. For example, infection disease status, autoimmune disease, and immune deficiency disease history are important confounding factors and are not provided from TCGA. To avoid the misunderstanding due to the model misspecification, analyses whose gc lambda > 5 or < 0.3 were removed. If gc lambda is > 1, we update gene-level chi-square statistics by dividing it over gc lambda and then calculate p-values. Then, false discovery rate (FDR) q-values were used to correct for multiple testing. In this paper, the gene-trait association with FDR q-values < 0.05 were identified as a significant finding.

### Estimation for proportion of traits variance explained by significant genes

We fit an analysis of variance from linear and logistic model to estimate the proportion of trait variants explained by top genes whose FDR q-values < 0.05. For each cancer type and TME trait, in addition to the confounders, we incorporated the somatic mutations and CNA of all genes identified in genome-wide association testing. The variance proportions explained by all top genes, including both somatic mutations and CNA, were summed up as an overall variance explained by somatic variants.

Since we incorporated genotypes of all top genes in the model, the collinearity between genes can cause inconsistency of p-values and effect sizes compared to a univariate analysis. Although the gene-level variance proportion should be interpreted carefully, the overall variance explained by somatic variants is still accurate.

### NRF2 gene signature curation

Three sets of NRF2 downstream genes are collected from 3 independent literatures. For Okazaki et al. 2020, 77 canonical NRF2 target genes identified by monitoring epigenetic profiling changes in NRF2 knockdown experiments from lung cancer cell line A549 are collected [37]. For Qian et al. 2015 and Namani et al. 2017, 593 genes and 477 genes that are downregulated by NRF2 knock down experiment from A549 cell line are collected respectively from associated literatures.[38,39]

### NRF2 signature score calculation

The gene signature score was calculated by using ssGSEA method from GSVA R package for each sample based on gene expression profile. For scRNA-seq specifically, pseudo bulk data aggregated from all cancer (malignant) cells are used. The median NRF2 signature score of all samples is used as cutoff to separate all samples into two groups NRF2$^{high}$ and NRF2$^{low}$ with each group has 22 samples. Fisher exact test is used to test enrichment of KEAP1 mutation within NRF2$^{high}$ samples(p.value = 0.00604).

For TCGA bulk RNA-seq a different cutoff was used based on the score distribution of KEAP1 mutated vs wild type to best separate the two populations. Based on Fig 4B, samples with NRF2 signature score > 1.25 are considered as NRF2$^{high}$ while others are considered as NRF2$^{low}$.

### Single cell RNA-seq data processing

Raw counts are downloaded for public datasets (GSE123902, GSE131907, and PRJNA591860) following standard Seurat procedure including data normalization, variable gene selection (Ngene = 2000), scaling, dimensionality reduction, and clustering using Seurat v4.1.0. All three datasets provide cell type annotation information which were used to identify malignant cells and immune cells separately.

To unify cell types across different studies, we re-annotated immune cells from each study. First, we extract immune cells from each dataset and used Harmony to remove potential batch effect by individual patient/sample. After integration, cells were plotted into a Uniform Manifold Approximation and Projection (UMAP) dimension based on reduction matrix by Harmony. Neighbor analysis was performed by invoking *FindNeighbors()* using Harmony reduction and the first 30 principle components. Clustering was then performed with *FindClusters()* at resolution 0.3.

For cell type annotation, we applied three different strategies: reference-mapping approach by Seurat [48], reference-based annotating by SingleR [49] and cell marker-based annotation

by SCINA [50]. For Seurat reference-mapping, we used the coupled PBMC dataset as the reference dataset following a standard procedure. Annotation results were then summarized at the level of cluster given by the dominant cell type. For SingleR annotation, we specifically used pure immune cells expression profile from Blueprint/ENCODE projects from celldex R package[49]. In SingleR, annotation was performed on the aggregated clusters. Cell markers that were used for SCINA annotation were provided. SCINA were invoked with parameter *max_iter = 100, convergence_n = 10, convergence_rate = 0.999, sensitivity_cutoff = 0.8*. SCINA annotation were also summarized at the level of cluster as mentioned above. Results given by three methods were combined and fed into a voting system. Considering all these three methods use different methodology for cell annotation, final cell cluster annotation was defined if two or more methods is able to predict the same annotation. If not, manually curation were performed to assign the cell type for that cluster.

## CellPhoneDB analysis

We used the CellphoneDB (CPDB) database to investigate difference in cell-to-cell communication in lung cancer. Ligand-receptor interaction scores were computed and tested using cellphonedb *method statistical_analysis*, executed on the tumor samples of either NRF2 Sig [high] or NRF2 Sig [low] samples. Differential ligand-receptor interaction was compared by calculating log2FC of ligand-receptor intensity between NRF2 Sig [high] and NRF2 Sig [low] samples. To study the interaction between myeloid cells and T cells, a user-specific custom database was generated by adding FCGR family and PD1 interaction.

## Differential gene expression testing

We used Seurat function *FindMarkers()* to call differential expression between NRF2 Sig [high] and NRF2 Sig [low] samples. Gene Set Enrichment Analysis (GSEA) was performed to study pathway activity in NRF2 Sig [high] samples.

## Survival analysis

We used "survival" package from R to do survival analysis with clinical data download from TCGA data portal. For single gene expression-based analysis, patients are separated into two groups by median expression. For NRF2 signature survival analysis, patients are separated as described previously.

## Clinical trial data re-analysis

We downloaded gene expression profile data (TPM) from GEO: GSE136961 and anti-PD-1 response data from the original paper [16]. Patients are separated into two groups as defined in the paper, group with durable benefits and group with non-durable benefits. Wilcoxon test was performed in R to compare the difference of NRF2 signature score between the two groups.

## Supporting information

**S1 Table. Summary of 74 TME traits in genome-wide association analyses.**
(CSV)

**S2 Table. Summary of 22 cancer types in genome-wide association analyses.**
(CSV)

**S3 Table. 451 significant gene-trait associations with q-values < 0.05 across 22 cancer types and 74 TME traits.** The analyses only include somatic mutations in coding region.
(CSV)

**S4 Table. Overall proportion of variance explained by somatic variants to TME traits.**
(CSV)

**S5 Table. Gene-level proportion of variance explained by somatic variants to TME traits.**
(CSV)

**S6 Table. Significant gene-trait associations with q-values < 0.05 across 22 cancer types and 74 TME traits.** The analyses include somatic mutations in both coding and non-coding regions.
(CSV)

**S7 Table. Significant gene-trait associations with q-values < 0.05 across 22 cancer types and 74 TME traits.** The analyses only include missense mutations.
(CSV)

**S8 Table. Significant gene-trait associations with q-values < 0.05 across 22 cancer types and 74 TME traits.** The analyses only include nonsense mutations.
(CSV)

**S9 Table. Significant gene-trait associations with q-values < 0.05 across 22 cancer types and 74 TME traits.** The analyses double weight driver mutations.
(CSV)

**S10 Table. Counts of somatic variants in different variant classes across 22 cancer types.**
(CSV)

**S1 Fig. Sensitivity analyses results of the significant associations between genes and TME traits.** In all the four panels, the x-axis corresponds to the analysis including only coding regions (denoted as Coding). The y-axis corresponds to the alternative analyses as below. (a) the somatic mutations in non-coding regions were additionally included (denoted as NonCoding); (b) driver somatic mutations were double-weighted (denoted as Driver); (c) only somatic mutations classified as Nonsense Mutation were included (denoted as Nonsense); (d) only somatic mutations classified as Missense Mutation were included (denoted as Missense). In our analysis, genes with fewer than 5 somatic mutation carriers were excluded for both coding and alternative analyses. Consequently, the sets of genes included in different analyses were not identical. For genes not included in either coding or alternative analyses, we set the corresponding p-value to 1.
(JPEG)

## Acknowledgments

This research is supported by High-performance Computing Platform of Peking University. Jun Li gives great suggestions for scRNA-seq data analysis.

## Author Contributions

**Conceptualization:** Xing Tang.

**Data curation:** Wenjian Bi, Zhiyu Xu, Xing Tang.

**Formal analysis:** Wenjian Bi, Zhiyu Xu, Peipei Zhang, Xing Tang.

**Funding acquisition:** Wenjian Bi.

**Investigation:** Wenjian Bi, Feng Liu, Zhi Xie, Hao Liu, Xiaotian Zhu, Wenge Zhong, Peipei Zhang, Xing Tang.

**Methodology:** Wenjian Bi, Zhiyu Xu.

**Project administration:** Wenjian Bi, Wenge Zhong, Peipei Zhang.

**Supervision:** Wenjian Bi, Wenge Zhong, Xing Tang.

**Validation:** Zhiyu Xu, Feng Liu, Zhi Xie, Hao Liu, Xiaotian Zhu, Wenge Zhong, Peipei Zhang, Xing Tang.

**Visualization:** Wenjian Bi, Zhiyu Xu, Peipei Zhang.

**Writing – original draft:** Wenjian Bi, Zhiyu Xu, Peipei Zhang, Xing Tang.

**Writing – review & editing:** Feng Liu, Zhi Xie, Hao Liu, Xiaotian Zhu, Wenge Zhong, Peipei Zhang, Xing Tang.

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
