## [Decision Letter · Decision Letter 0]

10 Oct 2023

Dear Dr Bi,

Thank you very much for submitting your Research Article entitled 'Genome-wide analyses reveal the contribution of somatic variants to the immune landscape of multiple cancer types' to PLOS Genetics.

The manuscript was fully evaluated at the editorial level and by independent peer reviewers. The reviewers appreciated the attention to an important problem, but raised some substantial concerns about the current manuscript. Based on the reviews, we will not be able to accept this version of the manuscript, but we would be willing to review a much-revised version. We cannot, of course, promise publication at that time.

If you decide to revise the manuscript for further consideration at PLOS Genetics, please aim to resubmit within the next 60 days, unless it will take extra time to address the concerns of the reviewers, in which case we would appreciate an expected resubmission date by email to plosgenetics@plos.org.

We are sorry that we cannot be more positive about your manuscript at this stage. Please do not hesitate to contact us if you have any concerns or questions.

Yours sincerely,

Peter Hammerman, MD, PhD

Academic Editor

PLOS Genetics

David Kwiatkowski

Section Editor

PLOS Genetics

Reviewer's Responses to Questions

**Comments to the Authors:**

Reviewer #1: In this study Bi et al. taking advantage of TCGA dataset to analyze the genome-wide association between genetic mutations and immune traits in the tumor microenvironment in different cancer types. They found that distinct gene mutations account for immune traits in different cancer types. In the end, they focused on the KEAP1 gene. The mutation of which is associated with NRF2 pathway activation, and consequently remodel the tumor immune microenvironment in lung adenocarcinoma. More importantly, the NRF2 activation status could serve as a better responsive biomarker for immune therapy. This is a very interesting study, and may have translational values as well. Here are a few minor concerns that can be addressed:

1. All the analysis and conclusions are drawn from bioinformatic analysis. It would be great to validate some key findings experimental. This is minor, as it would be very challenging to get patient samples to confirm gene mutation or signal pathway activation.

2. The study mainly focused on somatic mutations in coding regions. As non-coding regions (Long non-coding RNAs) also greatly contribute to TME remodeling. Would it be possible to include them as well?

3. The authors investigated the associations between mutations and TME traits. Would it be possible to classify types of gene mutations and TME traits? Such as deletion, nonsense mutation or missense mutation.

Reviewer #2: The authors performed genome-wide analysis and found associations between somatic mutations and immune landscape in various cancer types. In the case study of lung adenocarcinoma, the authors demonstrate KEAP1 mutations activate the NRF2 pathway and further lead to decreased T cell infiltration and increase T cell exhaustion. They also curated a NRF2 gene signature that shows better patient stratification in anti-PD1 therapy response than CD274 (PD-L1) alone. Overall, the study is clear and has made interesting findings, but definitely needs more work on emphasizing novelty and significance, rationalizing the study design, and polishing the figures (See details below).

Major comments:

• KEAP1 and NRF2 have been studied in NSCLC as well as other cancer types. It is not clear what discoveries have been made prior to this work, and how this work builds upon and extends beyond previous findings. Providing such context in the introduction helps establish the significance of the study.

• The impact of findings in the discussion needs to be strengthened. For example, authors highlighted that the curated NRF2 signature can serve as a better biomarker than PD-L1, but lacking the discussion about how this biomarker can be used practically in the clinical setting, considering RNA-seq is not routinely performed as a screening method.

• Are known driver genes and passenger genes treated differently in this study? Elaborate how the current method allows discovery of causal relationship between somatic mutations and immune landscape in TME. If not, provide discussion about the limitation of the method design.

• In general, figures need to be reworked. Legends need to include more details regarding what statistics are reported along the figures and how to interpret them. Figure arrangement, panel and font size consistency, and readability need to be improved.

Minor comments:

• The order of the figure panels should match the text. Specifically, figure 1b & c are LUAD validation and should go with figure 4c when the single-cell studies are talked about in the results section.

• The immune traits were transformed to quantitative or binary values depending on their distributions, which are categorized into three distinct patterns. More reasonings are needed here, e.g., what does raw trait of 0 mean in different situations? Are they specific to a subgroup of immune traits or cancer type?

• In figure 3a, would like to see other hits as well to show that KEAP1 is the ‘top gene’.

• In figure 4b, show statistical test comparing NRF2 signature scores of KEAP1 WT and mutated samples.

• Figure 5b panels and legends do not match. Bottom panel of figure 5a unclear, e.g. what are the lines and boxes, sizes of the box mean, etc. It is not clear what normalized expression is plotted in figure 5e heatmap and how it addresses the claims made in the text. Should it be correlation heatmap instead?

• FAM3C stratified patient groups in figure 6c do not show statistical differences.

• The x-axis timelines for survival plots in figure 6c, d and figure 7b seem suspiciously long (250 months). Please double check the analysis and plotting.

**Have all data underlying the figures and results presented in the manuscript been provided?**

Reviewer #1: Yes

Reviewer #2: Yes

PLOS authors have the option to publish the peer review history of their article (what does this mean?). If published, this will include your full peer review and any attached files.

Reviewer #1: No

Reviewer #2: No

---

## [Decision Letter · Decision Letter 1]

3 Jan 2024

Dear Dr Bi,

Thank you very much for submitting your Research Article entitled 'Genome-wide analyses reveal the contribution of somatic variants to the immune landscape of multiple cancer types' to PLOS Genetics.

The manuscript was fully evaluated at the editorial level and by independent peer reviewers. The reviewers appreciated the attention to an important topic but identified some concerns that we ask you address in a revised manuscript.

We therefore ask you to modify the manuscript according to the review recommendations. Your revisions should address the specific points made by each reviewer.

Yours sincerely,

Peter Hammerman, MD, PhD

Academic Editor

PLOS Genetics

David Kwiatkowski

Section Editor

PLOS Genetics

Reviewer's Responses to Questions

**Comments to the Authors:**

Reviewer #2: The authors addressed most of the comments very well.

Minor comment:

In the additional sensitivity analysis figure S1, the figure contents are somewhat confusing. The x, y-axes labels are supposed to be statistical significance from Coding and alternative analyses, respectively. However, for the genes that were only identified in one of the Coding or alternative analyses, the x,y values of the genes are the same (the significant p-value), making the points always fall on the x=y line, and therefore the xy axis definitions are not aligned. I found this design misleading. Is there a specific reason against plotting the p-values as they are? In addition, can you make some general comments on the characteristics (if any) of the genes identified only in the alternative analyses and why they are not of interest.

**Have all data underlying the figures and results presented in the manuscript been provided?**

Reviewer #2: Yes

PLOS authors have the option to publish the peer review history of their article (what does this mean?). If published, this will include your full peer review and any attached files.

Reviewer #2: No

---

## [Editor Report · Decision Letter 2]

9 Jan 2024

Dear Dr Bi,

We are pleased to inform you that your manuscript entitled "Genome-wide analyses reveal the contribution of somatic variants to the immune landscape of multiple cancer types" has been editorially accepted for publication in PLOS Genetics. Congratulations!

Yours sincerely,

Peter Hammerman, MD, PhD

Academic Editor

PLOS Genetics

David Kwiatkowski

Section Editor

PLOS Genetics

Comments from the reviewers (if applicable):

**Data Deposition**

http://datadryad.org/submit?journalID=pgenetics&manu=PGENETICS-D-23-01006R2

**Press Queries**

---

## [Editor Report · Acceptance letter]

16 Jan 2024

PGENETICS-D-23-01006R2 

Genome-wide analyses reveal the contribution of somatic variants to the immune landscape of multiple cancer types 

Dear Dr Bi, 

We are pleased to inform you that your manuscript entitled "Genome-wide analyses reveal the contribution of somatic variants to the immune landscape of multiple cancer types" has been formally accepted for publication in PLOS Genetics! Your manuscript is now with our production department and you will be notified of the publication date in due course.

With kind regards,

Anita Estes

PLOS Genetics

On behalf of:
